# [Proposal-ML] Efficient Deep Learning for Disaster Tweet Classification

**Isak Tønnesen, André Leonor, Guilherme Diogo**
Tsinghua University - Computer Science Department

## Abstract

Social media platforms have become crucial communication channels during emergencies and natural disasters, with Twitter emerging as a primary source of real-time situation awareness. While large-scale deep learning approaches have shown promise in disaster detection, there remains a critical need for efficient, deployable solutions that can operate within practical computational constraints. We propose a lightweight deep learning architecture for disaster tweet classification that combines pre-trained word embeddings with carefully engineered linguistic features. Our approach aims to achieve near state-of-the-art performance while requiring minimal computational resources, making it suitable for real-world deployment in resource-constrained environments. The system will be developed and evaluated using the Kaggle Natural Language Processing with Disaster Tweets dataset.

## 1 Background

During mass emergencies, time-sensitive information sharing becomes critical for public safety. Social media platforms, particularly Twitter, serve as vital emergency communication channels, enabling real-time disaster reporting by eyewitnesses. The challenge lies in efficiently processing this information stream to extract actionable intelligence. While large language models have demonstrated impressive results in disaster detection, their substantial computational requirements often make them impractical for real-world deployment, especially in resource-constrained environments where rapid response is crucial.

## 2 Problem Definition

Given a tweet $t$, we aim to determine whether it contains information about a real disaster ($f(t) \rightarrow \{0, 1\}$). Formally, our task is to develop a classification function $f$ that maps input text to binary labels while optimizing for both accuracy and computational efficiency:

$$f(t) =_{y \in \{0,1\}} P(y|t; \theta) \tag{1}$$

where $\theta$ represents our model parameters, constrained by computational resources $R$:

$$compute(\theta) \leq R \tag{2}$$

The challenge is to maximize classification accuracy while maintaining inference time below real-time requirements and memory usage within edge-device constraints.

38th Conference on Neural Information Processing Systems (NeurIPS 2024).

# 3  Related Work

Previous research has established strong foundations in social media disaster detection. Verma et al. [1] demonstrated that tweets containing situational awareness information exhibit specific linguistic characteristics, achieving over 80% accuracy using traditional NLP features. Klein et al. [2] proposed real-time clustering techniques for emergency event detection, showing that text-based features outperform metadata approaches.

Recent work by Maldonado et al. [3] demonstrated successful categorization of disaster tweets into four categories using NLTK-based classification, achieving 93.55% efficiency. Their work particularly highlighted the importance of efficient preprocessing and feature selection. Goswami and Raychaudhuri [4] explored decision tree-based approaches for disaster tweet classification, demonstrating that lightweight models can achieve comparable results to more complex architectures.

The gap between theoretical capabilities and practical implementation remains significant. While large-scale models show impressive results, their resource requirements make them impractical for many real-world scenarios, motivating our focus on efficient architectures.

# 4  Proposed Method

Our approach combines modern deep learning techniques with classical NLP features in a lightweight yet effective architecture:

## 4.1  Text Representation

We utilize pre-trained word embeddings (GloVe or FastText) for efficient text representation, complemented by carefully selected linguistic features proven effective in previous research. This provides rich semantic information while minimizing computational overhead.

## 4.2  Model Architecture

The core consists of:

- A bidirectional LSTM network (128 hidden units) for sequence processing
- Global pooling operations for efficient information combination
- A feed-forward neural network with dropout and batch normalization

## 4.3  Training Strategy

Our training approach emphasizes efficiency:

- Stratified k-fold cross-validation for robust evaluation
- Early stopping and gradual unfreezing of embedding layers
- Mixed precision training for reduced memory requirements
- Binary cross-entropy loss with Adam optimizer

# 5  Expected Results

We anticipate achieving results within 5-10% of state-of-the-art performance while requiring only a fraction of the computational resources. This performance-efficiency tradeoff represents a valuable contribution to the field, particularly for organizations with limited computational resources.

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
