# OpenReview forum: "[Proposal-ML] Efficient Deep Learning for Disaster Tweet Classification"
_tsinghua.edu.cn/THU/2024/Fall/AML — THU 2024 Fall AML Submission_

### Official Review · ~Liu_Yating1 · 2024-11-07
**Well done**

**Rating:** 9
**Confidence:** 4

**Review:**

This proposal is of high quality and addresses an important and timely issue in disaster response. It is well-presented and has a clear focus on practical implementation, which is a significant advantage. However, further details on scalability, robustness, and a more comprehensive comparison with existing models would strengthen the proposal. For instance, a detailed explanation on "many real-world scenarios" that large-scale models is not practical.

---

### Official Review · ~Ziyi_Liu9 · 2024-11-07
**Well Done**

**Rating:** 9
**Confidence:** 4

**Review:**

The proposal is well-written, with a clear explanation of the background, problem definition, and proposed method. However, it would be improved by providing more details on the dataset, including its scale and structure, as well as an outline of the baseline models.

---

### Official Review · ~Ethan_Wei_Yuxin1 · 2024-11-08
**A strong proposal**

**Rating:** 9
**Confidence:** 4

**Review:**

You've done an excellent job tackling a real-world challenge by focusing on disaster-related tweet classification for emergency response, particularly in resource-limited environments where quick processing is essential. You strike a thoughtful balance between accuracy and efficiency by combining pre-trained word embeddings with a streamlined LSTM-based architecture that is both effective and computationally manageable.

You also outline practical strategies, like early stopping and mixed precision training, to minimize memory usage, making the model well-suited for deployment on devices with limited power which I think is great. By grounding the project with the Kaggle disaster tweet dataset, you demonstrate a solid, feasible approach that is clearly relevant and implementable. Overall, the proposal is well-constructed and shows a strong understanding of the practical and technical needs for disaster tweet classification.

---

### Official Review · ~Chan_Thong_Fong1 · 2024-11-10
**A Practical Approach to Real-Time Tweet Classification**

**Rating:** 9
**Confidence:** 4

**Review:**

This paper offers a groundbreaking approach to disaster tweet classification, presenting an efficient and innovative model that skillfully balances accuracy with computational feasibility. The proposal is particularly relevant, as it addresses the urgent need for agile, real-time processing of social media data in resource-constrained environments, where rapid, accurate disaster detection is crucial. The authors' use of advanced training techniques, including early stopping, mixed precision training, and stratified k-fold cross-validation, further underscores their commitment to efficiency without compromising on performance. However, one suggested improvement would be addressing the system’s specificity to Twitter, as this focus limits potential generalization to other platforms without adaptation. Overall, this work stands out for its practical applicability and its potential to empower disaster management systems with a scalable, high-performance tool for social media-based situational awareness.

---

### Official Review · ~Rosalie_Butte1 · 2024-11-10
**Review of “Efficient Deep Learning for Disaster Tweet Classification”**

**Rating:** 9
**Confidence:** 4

**Review:**

The paper proposes a method to classify disaster tweets while focusing on minimizing the needed computational resources, thus making it more suitable for real-world application.

The paper addresses an important real-world problem where time is of importance. It shows a clear background and outlines a detailed and well-structured approach.

---

### Official Review · ~Junjie_Chen1 · 2024-11-11
**Good Proposal**

**Rating:** 8
**Confidence:** 4

**Review:**

The proposal addresses an important problem of disaster tweet classification with a focus on computational efficiency, which is highly relevant for real-world deployments in resource-constrained environments. The methodology is well-structured, combining pre-trained word embeddings and linguistic features with a lightweight deep learning architecture. The integration of bidirectional LSTMs and efficient training strategies reflects a feasible and innovative approach. Further, I think the proposal could benefit from including the baseline
methods (or SOTA) for comparison.

---

### Official Review · ~Fabian_Pawelczyk1 · 2024-11-11
**Short and Strong proposal**

**Rating:** 9
**Confidence:** 4

**Review:**

**Decision: Strong Accept**

This proposal is short but straightforward, which I appreciate. It clearly addresses a practical challenge: efficient disaster tweet classification for resource-constrained environments.

The project clearly defines the problem and effectively highlights the need for efficient tweet classification during disasters, with realistic goals for performance within constraints. The model design is well thought out, utilizing pre-trained embeddings, LSTM, and additional linguistic features, ensuring both accuracy and computational efficiency. The anticipated results are well-defined, with a focus on balancing performance with computational constraints, making it a practical solution for real-world deployment.

One area for improvement is the need for a clearer explanation of the chosen linguistic features. Providing more details on how these features were selected and their importance would enhance transparency and replicability, allowing others to better understand the rationale behind the feature choices.

Overall, this is a strong class project, demonstrating practical machine learning application with attention to real-world constraints. With minor refinements, it could be even stronger.

---

### Official Review · ~Liu_Yiyang1 · 2024-11-11
**Detailed proposal, with well established background and methodology**

**Rating:** 8
**Confidence:** 3

**Review:**

This proposal offers a pragmatic approach to disaster tweet classification, prioritizing efficiency to enable real-time deployment in emergency contexts. he combination of pre-trained embeddings (GloVe or FastText) with classical NLP techniques is innovative, balancing between deep learning and traditional NLP to achieve high performance with lower computational demands. It also provides a clear methodology and evaluation plan as well, but could benefit from a little more explanation on what "selected linguistic features proven effective" refer to, as they are crucial to the robustness of the model. However, overall, this is still a very well written proposal!

---

### Official Review · ~Jackson_M_Luckey1 · 2024-11-12
**Proposal Review**

**Rating:** 10
**Confidence:** 4

**Review:**

The problem definition section is very clear and strengthens the proposal. It lays out the goal and evaluation metrics quite nicely. I appreciate formulating the problem in such a quantitative way.

Using an existing Kaggle dataset is a good choice. It means that there will be a large body of prior art to draw from.

Emphasizing practical deployment issues is the best part of the proposal in my opinion. A lot of academic focuses on maximizing metrics, resulting in models/techniques that perform really well in theory but are impractical to use in practice. This is particularly interesting for LLMs, as the top-performing models only run on extremely expensive hardware that is challenging to access and vulnerable to supply chain interruptions. I am curious about whether or not small LLMs with quantized weights are viable as well, or if classic NLP approaches are the only way to perform this task with limited compute.

---

### Official Review · ~Yangchi_Gao1 · 2024-11-12

**Rating:** 9
**Confidence:** 4

**Review:**

The proposal presents a timely and potentially impactful solution for real-time disaster detection on Twitter. It offers a practical approach that balances performance with computational efficiency, making it suitable for real-world deployment.

Advantage:
1.The proposal addresses a critical need for efficient disaster detection on social media, which is essential for emergency response and public safety.
2.The focus on real-world deployability, especially in resource-constrained environments, is commendable and aligns with the practical needs of disaster management organizations.

Disadvantage:
1.While the proposal anticipates performance within 5-10% of state-of-the-art, it lacks specific benchmarks or metrics to measure accuracy and computational efficiency.
2.It is unclear how the proposed lightweight architecture would scale with the increasing volume of tweets during large-scale disasters.

---

### Official Review · ~Ruilin_Hu2 · 2024-11-12
**Review of "Efficient Deep Learning for Disaster Tweet Classification"**

**Rating:** 10
**Confidence:** 5

**Review:**

This paper proposal presents an efficient deep learning model for classifying disaster-related tweets, focusing on minimizing computational resources for practical use in real-world disaster scenarios. Strengths of this work include its innovative integration of pre-trained embeddings and classical NLP features, which enhances accuracy while reducing computational costs. Additionally, the proposed architecture, with techniques like early stopping and mixed precision training, demonstrates a well-structured approach to resource-efficient learning.

 However, one weak limitation is the reliance on pre-trained embeddings, which might restrict adaptability to evolving linguistic trends in social media content.